# Tumor Promoting Effect of BMP Signaling in Endometrial Cancer

**DOI:** 10.3390/ijms22157882

**Published:** 2021-07-23

**Authors:** Tomohiko Fukuda, Risa Fukuda, Kohei Miyazono, Carl-Henrik Heldin

**Affiliations:** 1Science for Life Laboratory, Department of Medical Biochemistry and Microbiology, Box 582, Uppsala University, SE-751 23 Uppsala, Sweden; risa.fukuda@imbim.uu.se (R.F.); miyazono@m.u-tokyo.ac.jp (K.M.); 2Department of Molecular Pathology, Graduate School of Medicine, The University of Tokyo, Tokyo 113-0033, Japan

**Keywords:** endometrial cancer, BMP, ACVR1, EMT, cancer stem cells

## Abstract

The effects of bone morphogenetic proteins (BMPs), members of the transforming growth factor-β (TGF-β) family, in endometrial cancer (EC) have yet to be determined. In this study, we analyzed the TCGA and MSK-IMPACT datasets and investigated the effects of BMP2 and of TWSG1, a BMP antagonist, on Ishikawa EC cells. Frequent *ACVR1* mutations and high mRNA expressions of BMP ligands and receptors were observed in EC patients of the TCGA and MSK-IMPACT datasets. Ishikawa cells secreted higher amounts of BMP2 compared with ovarian cancer cell lines. Exogenous BMP2 stimulation enhanced EC cell sphere formation via c-KIT induction. BMP2 also induced EMT of EC cells, and promoted migration by induction of SLUG. The BMP receptor kinase inhibitor LDN193189 augmented the growth inhibitory effects of carboplatin. Analyses of mRNAs of several BMP antagonists revealed that *TWSG1* mRNA was abundantly expressed in Ishikawa cells. TWSG1 suppressed BMP7-induced, but not BMP2-induced, EC cell sphere formation and migration. Our results suggest that BMP signaling promotes EC tumorigenesis, and that TWSG1 antagonizes BMP7 in EC. BMP signaling inhibitors, in combination with chemotherapy, might be useful in the treatment of EC patients.

## 1. Introduction

Endometrial cancer (EC) arises from uterine endometrial epithelium and invades into uterine myometrium. EC is the sixth most common cancer in women, and shows a rising incidence partly due to increasing obesity and longer life-span. EC patients have relatively good prognosis because they are often diagnosed at early stages with symptoms such as abnormal bleeding and lower abdominal pain, but still about 90,000 patients world-wide die from EC per year [1]. EC is divided into endometrioid carcinoma, which is the most common histological subtype with relatively good prognosis, and non-endometrioid carcinoma with worse prognosis [2]. Endometrial carcinosarcoma (ECS) is a rare histologic subtype of EC, which contains both carcinomatous and sarcomatous components, and causes around 16% of deaths due to malignancies of uterine corpus [3,4]. The sarcomatous component is considered to be derived from the carcinomatous component in most cases [4].

Transforming growth factor-β (TGF-β) is a major inducer of epithelial-mesenchymal transition (EMT) [5]. The TGF-β pathway has been reported to be activated in ECS [6], and TGF-β has tumor promoting as well as tumor suppressing effects in EC [7,8,9,10]. However, the effects of bone morphogenetic proteins (BMPs), members of the TGF-β family, in EC are not well known. BMP ligands exert their cellular effects by binding to and inducing complexes of type I and type II serine/threonine kinase receptors [11]. ACVRL1 (ALK1), ACVR1 (ALK2), BMPR1A (ALK3), and BMPR1B (ALK6) are classified as type I, and ACVR2A (ActRII), ACVR2B (ActRIIB), and BMPR2 (BMPRII) as type II, receptors [11]. After activation of the receptors, SMAD1/5/8 are phosphorylated and form complexes with SMAD4, which are translocated to the nucleus, where they regulate the transcription of several target genes, including *ID1* [11]. BMP ligands are also antagonized by several secreted proteins, such as Gremlin and TWSG1 [12].

DNA hypomethylation of the *BMP4* and *BMP7* genes was found to be associated with poor survival of EC patients [13]. Gremlin 2, an inhibitor of BMP signaling, was repressed in EC and inhibited EC cell growth in vitro [14]. However, the detailed effects of BMP signaling in EC cells have not been elucidated.

The goal of this study was to determine whether BMP signaling is tumor promoting or suppressing in EC cells, and to evaluate the effect of LDN193189, a BMP receptor kinase inhibitor on the growth and migration of EC cells in vitro. In addition, we investigated the effect of TWSG1, a modulator of BMP signaling expressed by EC cells, on the growth and migration of EC cells.

## 2. Results

### 2.1. BMP Signaling Is Activated in EC

The expression of mRNA for BMP ligands and receptors was found to be frequently increased in EC, as revealed by analysis of the TCGA EC database (Figure 1A). In addition, *ACVR1* mutations were more frequently observed in EC compared to other cancers, in both the TCGA and MSK-IMPACT datasets (Figure 1B). Around half of *ACVR1* mutations were R206H and G356D (Figure 1C), gain-of-function mutations commonly found in fibrodysplasia ossificans progressiva (FOP) and diffuse intrinsic pontine gliomas (DIPGs) [15]. Moreover, high expression of *BMP7* mRNA correlated with significantly lower survival of EC patients; the expression of *BMP2* mRNA also showed a correlation, albeit not significant, with poor EC patient survival (Figure 1D,E). We also performed survival analyses of other BMP ligands and receptors (Appendix A). However, there was no correlation between *ACVR1* mRNA expression and EC patient survival. To investigate the tumor promoting effect of BMP signaling in EC, we further performed in vitro experiments using Ishikawa EC cells, revealing expression of mRNA for all type I and type II BMP receptors, except *ACVRL1* (Figure 1F). *BMPR1A* mRNA was most abundantly expressed among the type I receptors, whereas *BMPR2* mRNA was most abundant among the type II receptors (Figure 1F). In addition, we found that Ishikawa cells secreted BMP2 at a higher level than OVSAHO and SKOV3 ovarian cancer cells, as determined by an ELISA (Figure 1G).

### 2.2. BMP2 Promotes EC Cell Stemness by c-KIT Induction

To determine whether Ishikawa cells responded to BMP stimulation, cells were stimulated by exogenous BMP2, and treated with or without LDN193189, a BMP type I receptor kinase inhibitor. BMP2 induced SMAD1/5/8 phosphorylation, which was inhibited in the presence of LDN193189 (Figure 2A). BMP2 also enhanced stemness of Ishikawa cells, as determined by sphere formation; LDN193189 inhibited the effect (Figure 2B). In accordance with this result, BMP2 increased the expressions of mRNA for the cancer stem cell markers *CD44* and *c-KIT* (Figure 2C). To determine whether c-KIT directly modulated stemness, it was overexpressed in Ishikawa cells (Figure 2D). c-KIT overexpression promoted sphere formation (Figure 2E), thus c-KIT enhanced stemness in Ishikawa cells. Moreover, the importance of c-KIT for the BMP2-induced stemness was investigated by knocking down c-KIT by two different siRNAs in Ishikawa cells; the knock-down efficiencies of the siRNAs were determined by qPCR (Appendix A). c-KIT knockdown neutralized BMP2-induced sphere formation (Figure 2F). In addition, inhibition of the tyrosine kinase activity of c-KIT by imatinib attenuated BMP2-induced sphere formation in Ishikawa cells (Figure 2G). These results suggest that BMP2 promotes EC stemness via c-KIT induction. To explore whether LDN193189 augments the growth inhibitory effects of carboplatin (CBDCA), a standard chemotherapeutic agent for EC patients, Ishikawa cells were treated with LDN193189 and CBDCA, alone and in combination. LDN193189 significantly potentiated the growth inhibitory effect of CBDCA, as determined by an MTS assay (Figure 2H). LDN193189 also augmented the inhibitory effect of CBDCA on sphere formation, in the absence or presence of BMP2 stimulation (Figure 2I, Appendix A).

### 2.3. BMP2 Induces EMT of EC Cells

We investigated the effect of BMP signaling on downstream genes in Ishikawa cells, and found that BMP2 induced *ID1*, *SNAIL* and *SLUG* mRNA in a time-dependent manner (Figure 3A). *ID1* induction was sustained till 72 h, whereas *SNAIL* and *SLUG* induction peaked rapidly at 2 h after BMP2 stimulation (Figure 3A). Since SNAIL and SLUG are EMT transcription factors, we investigated the expression of EMT markers by immunoblotting. The epithelial marker E-cadherin was suppressed by BMP2, whereas the mesenchymal markers N-cadherin and vimentin were enhanced in a time-dependent manner (Figure 3B); these effects were neutralized by treatment with LDN193189 (Figure 3C). E-cadherin attenuation and vimentin induction were also confirmed by immunofluorescent staining (Figure 3D). These results support the notion that BMP2 induces EMT in Ishikawa cells.

### 2.4. BMP2 Enhances EC Cell Migration via SLUG Induction

As BMP2 induced EMT in Ishikawa cells (Figure 3), we investigated whether BMP2 also enhanced cell migration. Using a scratch assay, we observed that BMP2 enhanced Ishikawa cell migration, and that LDN193189 reversed the effect (Figure 4A). Since SNAIL and SLUG were induced after BMP2 stimulation of cells (Figure 3A), we knocked down SNAIL or SLUG by siRNAs. SNAIL knockdown had no effect on BMP2-induced cell migration (Figure 3B), whereas SLUG knockdown inhibited BMP2-induced cell migration (Figure 3C). The induction of SLUG by BMP2 and its suppression by siRNAs was confirmed by qPCR (Appendix A). Moreover, SLUG knockdown reversed the suppression of E-cadherin induced by BMP2 and slightly attenuated the expressions of the mesenchymal markers N-cadherin and vimentin (Figure 3D). These results suggest that BMP2 enhances migration and EMT of EC cells in a SLUG-dependent manner.

### 2.5. TWSG1 Antagonizes BMP7 in EC Cells

We explored the possibility that BMP antagonists affected BMP signaling in Ishikawa cells. First, we assessed mRNA expressions of ten BMP antagonists in Ishikawa cells; *TWSG1* mRNA was most abundantly expressed (Figure 5A). However, according to TNM plot, TWSG1 mRNA expression was significantly decreased in EC compared with normal endometrium (Figure 5B). Expression of the TWSG1 protein was detected both in cell lysates and cultured medium (Figure 5C). Exogenous TWSG1 suppressed BMP7-induced, but not BMP2-induced, SMAD1/5/8 phosphorylation (Figure 5D), and expressions of *ID1* (Figure 5E) and *SLUG* (Figure 5F) mRNA. Consistent with these results, TWSG1 inhibited BMP7-induced, but not BMP2-induced, sphere formation (Figure 5G) and cell migration (Figure 5H) of Ishikawa cells. These results show that TWSG1 antagonizes BMP7 in EC cells.

## 3. Discussion

BMP signaling has tumor promoting or suppressing effects depending on the type of tumor [16]. In this study, we demonstrate that BMP signaling has tumor promoting effects on EC, and that both LDN193189, a BMP type I receptor kinase inhibitor, and to some extent TWSG1, reverse these effects on EC cells. LDN193189 also augments the growth inhibitory effects of CBDCA.

*ACVR1*, which we found to be expressed in Ishikawa cells, is frequently mutated in EC patients. *ACVR1* R206H and G356D, common causative mutations in FOP and DIPGs [15], were also found in EC. Thus, *ACVR1* gain-of-function mutations most likely promote endometrial carcinogenesis through hyper-activation of BMP signaling.

In addition, mRNAs of BMP ligands and receptors are over-expressed in EC. Overexpression of BMP ligands and receptors is also found in ovarian cancer [17]. Furthermore, high *BMPR2* mRNA expression correlates with poor prognosis in ovarian cancer [17]. These results suggest common tumor-promoting function of BMP signaling in EC and ovarian cancer patients.

BMP2 significantly enhanced EC cell sphere formation and migration. BMP2 binds with high affinity to BMPR2 and the type I receptor BMPR1A [11], which were both expressed at high levels in Ishikawa cells. In accordance with this, BMP2 clearly induced SMAD1/5/8 phosphorylation and *ID1* expression in Ishikawa cells; LDN193189 inhibited these effects. We previously reported that BMP2 also increased sphere formation and migration of ovarian cancer cells [17]. Considering that the expression patterns of BMP receptors were similar between EC and ovarian cancer cells [17], they may share a common response to BMP2 stimulation.

We found that BMP2 stimulation enhanced EC cell stemness in a c-KIT-dependent manner. c-KIT is a receptor tyrosine kinase, the ligand of which is a stem cell factor (SCF). Our finding that SCF/c-KIT signaling enhanced EC stemness is consistent with a previous report [18]. Moreover, sarcomatous components of ECS showed positive c-KIT staining [19,20]. A correlation between c-KIT expression and poor prognosis was also reported in EC patients [21].

We demonstrated that BMP2 induces EMT and migration of EC cells, including suppression of E-cadherin expression in a SLUG-dependent manner. Consistent with our observations, E-cadherin suppression has been shown to augment endometrial epithelial cell migration [22]. Since high expression of SLUG was associated with recurrence and poor survival in EC [23], SLUG induction by BMP signaling may be important for EC progression.

TWSG1 has been found to act both as an activator and an inhibitor of BMP signaling depending on the type of tissues [24,25]. We found that TWSG1 antagonized BMP7, but not BMP2, suggesting an interaction between TWSG1 and BMP7 [26]. The *TWSG1* mRNA expression in EC was found to be significantly lower than in normal endometrium, consistent with a tumor suppressive role of TWSG1 in EC.

Our study has several limitations. Whereas BMP signaling has clear tumor promoting roles in vitro, further evaluation of BMP signaling in vivo is needed. Although *ACVR1* gain-of-function mutation was found in EC patients, its importance in endometrial carcinogenesis was not determined. Further studies will be needed to elucidate the possible role of *ACVR1* mutations in EC development.

## 4. Materials and Methods

### 4.1. Chemicals and Antibodies

Imatinib and carboplatin were purchased from Sigma-Aldrich (St. Louis, MO, USA) and dissolved in water. LDN193189 was obtained from Yoshinobu Hashizume (RIKEN, Saitama, Japan), dissolved in DMSO (Sigma-Aldrich). BMP2, BMP7, and TWSG1 were purchased from R&D Systems (Minneapolis, MN, USA). Antibodies used in the present study are listed in Appendix A.

### 4.2. Bioinformatic Analysis

Gene expression profiles of EC patient samples were acquired from the cBioPortal (TCGA pancancer atlas studies and the MSK-IMPACT clinical sequencing cohort) [27,28]. We further analyzed the TCGA uterine corpus endometrial carcinoma dataset, which contained mutation data of 248 patients and RNA-Seq data of 177 patients by submitting a query to cBioPortal [27,29,30]. RNA expression cutoff Z score was adjusted to 2.0. Survival analyses were performed using RNA-Seq data of KM plotter, which contained 542 EC patients [31]. Patients were divided into two groups, above and below median expression, respectively. We also compared *TWSG1* mRNA expressions of EC and non-cancerous endometrium by submitting a query to TNM plot with selection of RNA-Seq data [32].

### 4.3. Cell Culture

Ishikawa EC cells, and SKOV3 and OVSAHO ovarian cancer cells, were gifts from Katsutoshi Oda (The University of Tokyo, Japan). The cells were cultured in DMEM or RPMI with 10% fetal bovine serum (FBS) at 37 °C in a humidified incubator with 5% CO_2_. The cell lines were regularly tested for the absence of mycoplasma.

### 4.4. RNA Extraction and Real-Time PCR

RNA extraction, cDNA synthesis, and real-time RT-PCR fluorescence detection were performed as previously described [17]. Primers for each gene are listed in Appendix A. The threshold cycle number (Ct) for each sample was determined in triplicate. The Ct values were normalized against *GAPDH*.

### 4.5. ELISA

Ishikawa, OVSAHO, and SKOV3 cells were cultured to confluence in complete medium. After washing with PBS, cells were incubated in serum-free medium for an additional 24 h. After passing through a 0.45 μm syringe filter, BMP2 was quantified by an ELISA kit from Invitrogen (Carlsbad, CA, USA), according to the manufacturer’s instructions.

### 4.6. Immunoblotting

Ishikawa cells were harvested and soluble protein was extracted, followed by immunoblotting using the indicated antibodies, as previously described [33]. To detect secreted proteins, Ishikawa cells were cultured in serum-free medium for 24 h. Cultured medium was collected and centrifuged at 3000 rpm for 5 min, after which 4 × volumes of acetone were added to the supernatants. After the samples had been kept at −20 °C overnight, they were centrifuged at 10,000 rpm for 10 min, whereafter pellets were dissolved in RIPA buffer. Proteins were detected using a BioRad immunoblotting system (BioRad, Hercules, CA, USA) with the Immobilon Western Chemiluminescent HRP substrate (Millipore, Burlington, MA, USA). α-tubulin was used as an internal control.

### 4.7. Sphere Formation Assay

Ishikawa cells (1 × 10^4^/well) were seeded into DMEM/F12 medium supplemented with 20 ng/mL EGF and 10 ng/mL bFGF from Sigma-Aldrich in 96-well Costar ultra-low attachment plates (Corning, NY, USA) and incubated with indicated reagents for 8 days. Total sphere numbers (diameters > 50 μm) per well were counted using a microscope.

### 4.8. Gene Silencing and Plasmid Transfection

Ishikawa cells were cultured for 24 h before gene silencing and plasmid transfection. Small interfering RNA (siRNA) for *c-KIT* (HSS105820, HSS105821), *SLUG* (HSS109993, HS109995) (Stealth RNAi siRNA from Invitrogen) and *SNAIL* (sc-38399; Santa Cruz Biotechnology, CA, USA), were used. Gene silencing was performed with Lipofectamine RNAiMAX transfection reagent (Invitrogen) according to the manufacturer’s instructions. Negative controls (siNC) were from the Stealth RNAi siRNA Negative Control Kit (Invitrogen). *c-KIT* expression plasmid was purchased from Origene (SC120061; Rockville, MD, USA) and transfected into Ishikawa cells using Lipofectamine 2000 transfection reagent (Invitrogen). pcDNA 3.0 (Invitrogen) was used as a control (CT). After 48 h incubation with siRNA or plasmid, cells were subjected to further experiments.

### 4.9. Immunofluorescence

Cells were treated with indicated reagents and fixed, as previously described [33]. Cells were permeabilized in 0.2% Triton X-100 for 10 min prior to blocking by incubation in 6% bovine serum albumin (BSA) for 30 min. Cells were incubated with primary antibodies against E-cadherin and vimentin at 4 °C overnight, followed by incubation with a secondary antibody, Alexa Fluor 488 Goat anti-Rabbit IgG, for 1 h at room temperature. Nuclei were counterstained with ProLong Gold Antifade Mountant with DAPI (Invitrogen). The cells were analyzed using a confocal fluorescence microscope (Axio Imager.M2; Carl Zeiss, Oberkochen, Germany).

### 4.10. Scratch Assay

Cell migration was assessed by a cell culture scratch assay. Ishikawa cells were cultured to confluence in 6-well plates. After scratching by a 10 μL pipette tip, cells were washed with PBS and incubated with medium containing 3% FBS in the presence of indicated reagents, for an additional 48 h. Images were taken at 0 and 48 h after the scratch. Cell motility was determined by measuring the width of the gap at 0 and 48 h.

### 4.11. MTS Assay

Cells were seeded into 96-well plates (3 × 10^3^ cells/well) and incubated with indicated reagents. After adding the tetrazolium salt MTS (CellTiter 96 AQueous One Solution; Promega, Madison, WI, USA) to each well, the absorbance at 450 nm was monitored by using the EnSpire multimode plate reader (PerkinElmer, Waltham, MA, USA). Cell numbers were normalized relative to the absorbance of cells treated with DMSO alone.

### 4.12. Statistical Analysis

Data are presented as the mean ± SE. The experiments were repeated at least three times, and representative data were shown from multiple experiments. The significance of differences between three or more samples was analyzed by one-way ANOVA with Tukey–Kramer test, whereas the significance between two samples was analyzed by two-tailed Student’s *t*-test. A *p*-value < 0.05 was considered statistically significant.

## 5. Conclusions

In conclusion, we have demonstrated tumor promoting effects of BMP signaling in EC cells, by induction of cancer stemness, EMT, and migration. We also showed that both the BMP type I receptor kinase inhibitor LDN193189 and TWSG1 have tumor suppressing effects via inhibition of BMP signaling. Given that BMP signaling is frequently activated in EC patients, combinational treatment with CBDCA and a BMP signaling inhibitor could be beneficial in the treatment of EC patients.

## Figures and Tables

**Figure 1 ijms-22-07882-f001:**
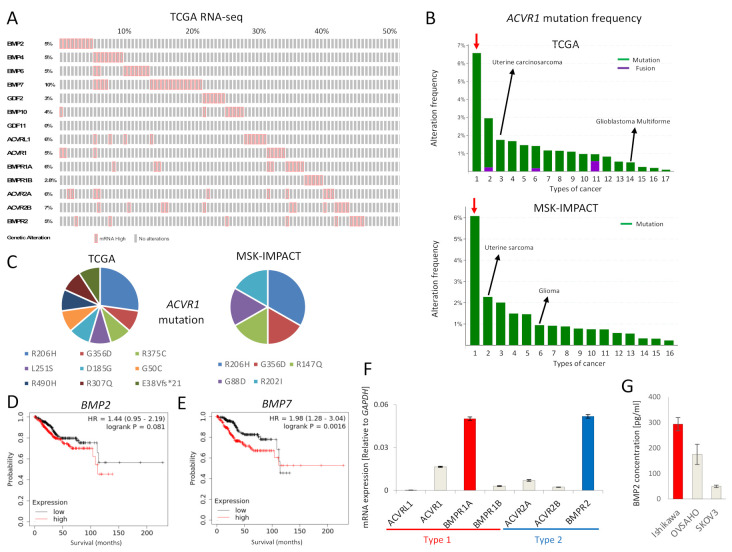
BMP signaling is activated in EC. (**A**) mRNAs for BMP ligands or receptors are over-expressed in EC. RNA-seq of the TCGA endometrial cancer dataset containing 177 EC tumors was analyzed via cBioPortal. RNA expression cutoff Z score was adjusted to 2.0. The results of 90 tumors are shown. (**B**) *ACVR1* is more frequently mutated in EC compared to other cancers. *ACVR1* mutation frequency of the TCGA pancancer atlas studies and of the MSK-IMPACT clinical sequencing cohort was assessed by cBioPortal. Red arrows point to EC. Lanes 1 to 17 (in the top panel); endometrial carcinoma, skin cutaneous melanoma, uterine carcinosarcoma, colorectal adenocarcinoma, bladder urothelial carcinoma, lung adenocarcinoma, mesothelioma, stomach adenocarcinoma, adrenocortical carcinoma, head and neck squamous cell carcinoma, ovarian serous cystadenocarcinoma, lung squamous cell carcinoma, liver hepatocellular carcinoma, glioblastoma multiforme, kidney renal clear cell carcinoma, brain lower grade glioma and breast invasive carcinoma, Lanes 1 to 16 (in the bottom panel); endometrial cancer, uterine sarcoma, melanoma, mesothelioma, cancer of unknown primary, glioma, esophagogastric cancer, colorectal cancer, hepatobiliary cancer, head and neck cancer, mature B-cell neoplasms, germ cell tumor, non-small cell lung cancer, renal cell carcinoma, bladder cancer and breast cancer. (**C**) Details of *ACVR1* mutations found in EC are shown. Eleven cases of the TCGA dataset (out of 244 cases) and six cases of the MSK-IMPACT dataset (out of 113 cases) had the *ACVR1* mutations indicated. (**D**,**E**) Overall survival was analyzed using RNA-Seq data of KM plotter, which contained 542 EC patients. Patients were divided into two groups, i.e., above or below median mRNA expression. The effects of expression of *BMP2* (**D**) and *BMP7* (**E**) on the survival of EC patients, are shown. (**F**) mRNA expression levels of BMP receptors in Ishikawa EC cells, as determined by qRT-PCR and normalized relative to *GAPDH*. (**G**) BMP2 secretion by Ishikawa cells, and by OVSAHO and SKOV3 ovarian cancer cells for comparison. Confluent cell cultures were incubated in serum-free medium for 24 h; thereafter, the conditioned medium was analyzed for BMP2 by an ELISA. BMP2 concentration was normalized to 1 mg total protein in lysates. The results in panel F and G are shown as the mean ± SE.

**Figure 2 ijms-22-07882-f002:**
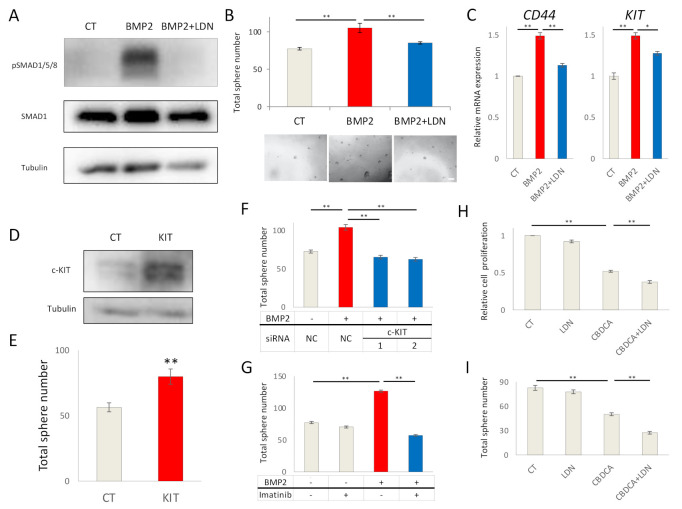
BMP2 promotes EC cell stemness by c-KIT induction. (**A**) BMP2 stimulation induces SMAD1/5/8 phosphorylation in Ishikawa cells. Cells were treated in the absence (CT) or presence of 20 ng/mL BMP2 and 200 nM LDN193189 (LDN) for 24 h. α-tubulin was used as an internal control. (**B**) BMP2 induces stemness of Ishikawa cells, as determined by a sphere formation assay. Cells were cultured with stem cell medium containing 20 ng/mL BMP2 and 200 nM LDN in 96-well ultra-low attachment plates for eight days; thereafter, sphere numbers per well were counted using a microscope. Images of spheres are shown at the bottom of the graphs. Scale bar = 200 µm. (**C**) BMP2 induces expression of *CD44* and *c-KIT* mRNA in Ishikawa cells. Cells were treated with PBS (CT), BMP2 (20 ng/mL) or LDN, alone or in combination, for 72 h. mRNA expression was determined by RT-PCR and is shown as fold change relative to control (CT). (**D**) c-KIT expression was quantified by immunoblots in Ishikawa cells 72 h after transfection with empty vector (CT) or c-KIT (KIT) plasmids. α-tubulin was used as an internal control. (**E**) Overexpression of c-KIT by transfection induces stemness of Ishikawa cells, as determined by a sphere formation assay. (**F**,**G**) BMP2-induced stemness of Ishikawa cells is dependent on c-Kit. Cells were transfected with siNC, siKIT-1, or siKIT-2 for 48 h; thereafter, cells were cultured for an additional eight days in the presence and absence of 20 ng/mL BMP2 (**F**), or incubated in the presence and absence of 20 ng/mL BMP2 and 10 µM imatinib (**G**). Cancer stemness was determined by the formation of spheres. (**H**,**I**) Ishikawa cells were incubated in the absence (CT) or presence of 200 nM LDN and 500 µM carboplatin (CBDCA). Cell proliferation was determined after 72 h by an MTS assay, and is expressed relative to CT (**H**), and stemness by a sphere formation assay after eight days (**I**). The results in panels B, C, E–I are shown as the mean ± SE. * *p*-value < 0.05, ** *p*-value < 0.01.

**Figure 3 ijms-22-07882-f003:**
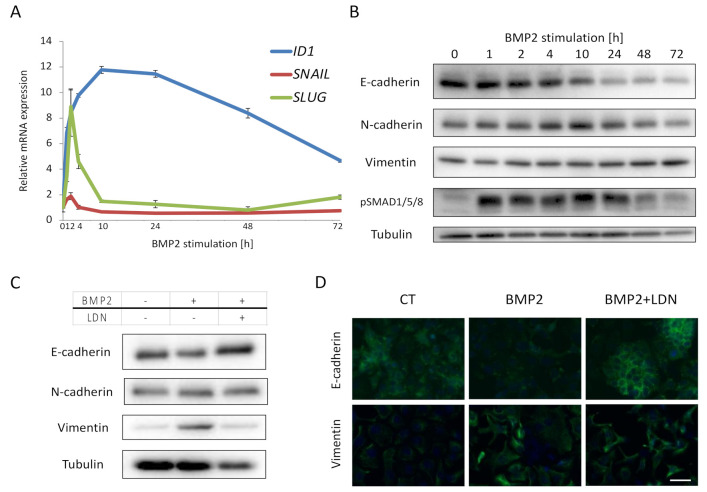
BMP2 induces EMT of EC cells. (**A**,**B**) Ishikawa cells were cultured in serum-free medium overnight and treated with 20 ng/mL BMP2 for the indicated time periods. Expression of *SNAIL*, *SLUG*, and *ID1* mRNA was analyzed by qRT-PCR and normalized relative to 0 h (**A**), and expression E-cadherin, N-cadherin, vimentin and phospho-SMAD1/5/8 was analyzed by immunoblots, using α-tubulin as a loading control (**B**). (**C**,**D**) Ishikawa cells were incubated in the absence (CT) or presence of 20 ng/mL BMP2 and 200 nM LDN, in 1% FBS-containing medium for 48 h, and then subjected to immunoblotting (**C**) and immunofluorescent staining (**D**) for EMT markers. Scale bar = 10 µm. The results in panel A are shown as the mean ± SE.

**Figure 4 ijms-22-07882-f004:**
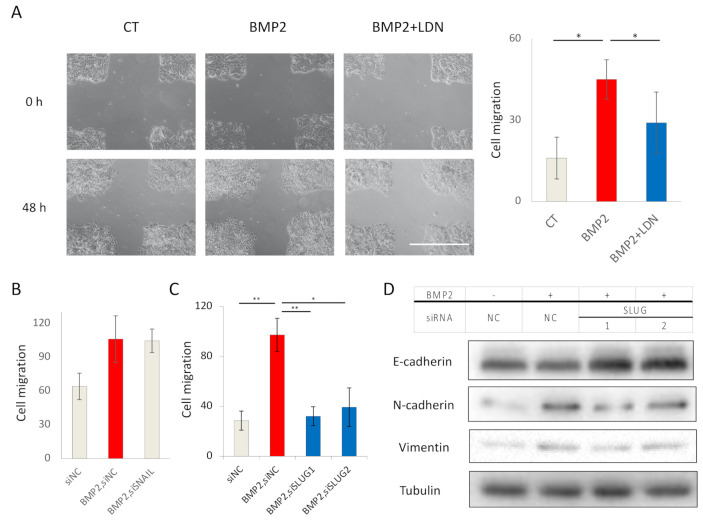
BMP2 enhances EC cell migration via SLUG induction. (**A**) Migration of Ishikawa cells was evaluated by a scratch assay. Confluent cell cultures were scratched by a 10 µL pipette tip and incubated in the absence (CT) or presence of 20 ng/mL BMP2 and 200 nM LDN193189 (LDN) in 3% FBS-containing medium for 48 h. Cell motility was determined by measuring the gaps between the cell sheets at 0 and 48 h. Scale bar = 100 µm. (**B**–**D**) BMP2-induced EC cell migration is dependent on SLUG, but not on SNAIL. Ishikawa cells transfected with siNC, siSNAIL (**B**), or siSLUG-1 or siSLUG-2 (**C**) for 48 h, were incubated in the absence (CT) and presence of 20 ng/mL BMP2 in 3% FBS-containing medium for an additional 48 h. Cell migration was analyzed by scratch assays (**B**,**C**), and expression of EMT markers was determined by immunoblotting (**D**). The results in panels A, B, and C are shown as the mean ± SE. * *p*-value < 0.05, ** *p*-value < 0.01.

**Figure 5 ijms-22-07882-f005:**
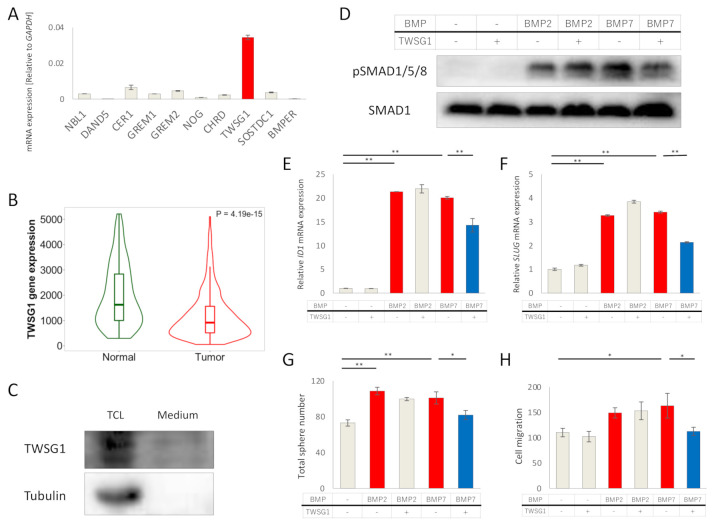
TWSG1 antagonizes BMP7 in EC cells. (**A**) *TWSG1* mRNA is abundantly expressed in Ishikawa cells. mRNA expression of ten BMP antagonists was determined by qRT-PCR and normalized relative to *GAPDH*. (**B**) *TWSG1* mRNA expressions of EC and non-cancerous endometrium were analyzed by submitting a query to TNM plot with selection of RNA Seq data. (**C**) Ishikawa cells secrete TWSG1. TWSG1 protein was detected by immunoblotting in both total cell lysates (TCL) and cultured medium (Medium) of Ishikawa cells. (**D**) TWSG1 inhibits BMP7-induced SMAD1/5/8 phosphorylation. Ishikawa cells were cultured overnight in serum-free medium, and then treated with or without 20 ng/mL BMP2, 50 ng/mL BMP7, and 1000 ng/mL TWSG1 for 3 h. Cell lysates were subjected to immunoblotting for P-SMAD1/5/8 and SMAD1. (**E**,**F**) TWSG1 suppresses BMP7-induced ID1 and SLUG expression. RNA was extracted from Ishikawa cells cultured under the same conditions as (**D**). *ID1* (**E**) and *SLUG* (**F**) mRNA expression was evaluated with qRT-PCR. mRNA expression was normalized relative to no stimulation. (**G**) TWSG1 decreases BMP7-enhanced sphere formation. Ishikawa cells were incubated with or without 20 ng/mL BMP2, 50 ng/mL BMP7 and 1000 ng/mL TWSG1 for eight days, where spheres were later counted using a microscope. (**H**) TWSG1 suppresses BMP7-induced EC cell migration. Ishikawa cell cultures were subjected to a scratch, after which cells were incubated in the absence or presence of 20 ng/mL BMP2, 50 ng/mL BMP7 and 1000 ng/mL TWSG1 in 3% FBS-containing medium; after 48 h, the widths of the scratches were determined. The results in panels A, E–H are shown as the mean ± SE. * *p*-value < 0.05, ** *p*-value < 0.01.

## Data Availability

Gene expression profiles of EC patient samples were acquired from the cBioPortal (https://www.cbioportal.org/ (accessed on 22 November 2020)). Survival analyses were performed using RNA-Seq data of KM plotter (http://kmplot.com/analysis/ (accessed on 5 May 2021)). *TWSG1* mRNA expressions of EC and non-cancerous endometrium were obtained from TNM plot (https://www.tnmplot.com/ (accessed on 5 May 2021)).

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
