# Peer review of "Tumor Promoting Effect of BMP Signaling in Endometrial Cancer"

_ijms, 2021, doi:10.3390/ijms22157882_

Round 1

Reviewer 1 Report

Fukuda et al. investigates BMP signalling in endometrial cancer. The manuscript provides new information on how BMP2 effects tumour progression, in particular that BMP2 promotes EMT and enhances stem cell characteristics in an endometrial cancer cell line.

Overall, it is well written and the methodology appropriate, apart from exceptions noted below. However, the study lacks a clear focus and doesn’t follow any logical progression. The overall aim is to investigate clinical and in vitro aspects of BMP signalling in EC. But results are only shown for a few selected components of the pathway in each figure. This lack of focus detracts from the manuscript considerably. I suggest the authors consider revising the  aims and hypotheses given that the majority of findings involve BMP2. The clinical analysis could be expanded to include other BMPs and receptors, and then a clear rationale provided for focussing on BMP2 in subsequent figures.

Points to address:

  1. The proposed focus of the paper is BMP signaling, however the in vitro data mostly investigates effects of BMP2. Were BMP4 and 8 also associated with clinical outcome? Why was BMP2 selected for further in vitro investigation and not BMP4, 7 or 8? BMP7 might have been more relevant given that tumours were more likely to have higher BMP7 mRNA expression in EC patients (fig 1) and BMP7 was associated with poor prognosis in the KM analysis, whereas BMP2 was not significant. Similarly, mutation analysis is only shown for one of the BMP receptors, ACVR1. Were the other BMP receptors assessed and if not, why? Is expression associated with EC outcome? The last figure of the results shows that TWSG1 antagonizes BMP7 in EC cells. It’s not clear why BMP7 is investigated here when the previous in vitro analysis only involves BMP2.
  2. Only one endometrial cancer cell line is investigated. Why was this cell line selected? It would be helpful to have at least some of the results confirmed in a second cell line. This is particularly relevant for the migrations studies as Ishikawa cells have low migratory properties.
  3. In Figure 2, LDN193189 augments the growth inhibitory effects of carboplatin (CBDCA). Although LDN193189 reverses the tumor promoting effects induced by exogenous BMP2, LDN193189 is not specific to BMP2 mediated Smad4/5/8 activation, other BMPs, such as BMP4, mediated Smad4/5/8 activation can also be inhibited by this antagonist. Thus, in figure 2H and I, observed effects may not only due to inhibition of BMP2- this needs to be clarified. Do Ishikawa cells express/secrete other BMP members?
  4. Title: Tumor promoting effect of BMP signaling in endometrial cancer. The data presented does not support that BMPs promote tumour growth as in vivo studies were not included. Consider changing the title to reflect the key outcomes.
  5. L33 “an increasing incidence partly due to increased incidence of obesity and increased life-span.” Consider re-wording this sentence.
  6. L74: qPCR in one cell line with only selected genes from the family does not ‘validate’ the clinical findings. Please correct this sentence.
  7. Figure 2 legend- were qPCR results normalised to housekeeping genes? This needs to be stated. The authors could also consider using CTRL for control in the qPCR graphs instead of CT, so as not to be confused with cycle threshold Ct (e.g. ‘shown as fold change relative to CT’ in the legend).
  8. There is no indication of the vehicle control used for experiments involving Imatinib LDN193189 and carboplatin i.e. what are the stocks dissolved in and were these used as vehicle control (where applicable).
  9. L337 ‘The significance of differences between more than three samples was analyzed by one-way ANOVA’ – should this read ‘three or more samples’?
  10. How were the different doses of BMP2 (20ng/ml) and BMP7 (50 ng/ml) selected?
  11. How were the ELISA results normalised? Cell number at confluency varies across different cell types so the experiments should be normalised e.g. to total protein in lysates

Author Response

Details of response to reviewers:

 We highly appreciate the editor and the reviewers for giving us the opportunity to revise our manuscript. We have repeated the comments of the reviewers and have responded to the comments. The page and line numbers refer to our revised manuscript.

Reviewer 1

Comments to the Author

  1. The proposed focus of the paper is BMP signaling, however the in vitrodata mostly investigates effects of BMP2. Were BMP4 and 8 also associated with clinical outcome? Why was BMP2 selected for further in vitroinvestigation and not BMP4, 7 or 8? BMP7 might have been more relevant given that tumours were more likely to have higher BMP7 mRNA expression in EC patients (fig 1) and BMP7 was associated with poor prognosis in the KM analysis, whereas BMP2 was not significant. Similarly, mutation analysis is only shown for one of the BMP receptors, ACVR1. Were the other BMP receptors assessed and if not, why? Is expression associated with EC outcome? The last figure of the results shows that TWSG1 antagonizes BMP7 in EC cells. It’s not clear why BMP7 is investigated here when the previous in vitro analysis only involves BMP2.

Response

We apologize that we showed only survival analyses of BMP2 and BMP7. We further performed survival analyses of other BMP ligands and receptors and found that high BMP8B and low BMPR1B mRNA expression also correlated with poor prognosis. We added this result to the supplementary figure. With regard to mutation analysis, we noticed that other BMP receptors were also mutated in EC to lesser extent than ACVR1. This may be due to POLE mutation, which leads to high mutation frequency as described in TCGA endometrial cancer’s manuscript. However, only ACVR1 mutation can be detected by clinical oncopanels including MSK-IMPACT and significance of several ACVR1 mutations have been elucidated. Therefore we excluded mutation analyses of other BMP receptors.

We selected BMP2 because Ishikawa cells had relatively high BMPR1A mRNA expression among type1 receptors. BMP2 and BMP4 are known to bind to BMPR1A with high affinity compared with other BMP ligands. But we also investigated the effect of BMP7 as shown in Figure 5. According to our result, BMP2 is a more potent ligand than BMP7 at least in Ishikawa cells.

According to literature, TWSG1 has been shown to interact with BMP2/7 among BMP ligands. We first investigated the combination of BMP2 and TWSG1. Since the combination had no effect, we switched BMP2 to BMP7.

Please see supplemental Figure S1 and page 2 lines 78–80 of the revised manuscript.

  1. Only one endometrial cancer cell line is investigated. Why was this cell line selected? It would be helpful to have at least some of the results confirmed in a second cell line. This is particularly relevant for the migrations studies as Ishikawa cells have low migratory properties.

Response

We selected the Ishikawa EC cell line because it is classified as a typical type1 EC cell line with retaining hormone sensitivity. We completely agree that we should repeat same experiments in another EC cell line if possible. However, Ishikawa cell line is the only EC cell line which we have in our laboratories and it is difficult to get another cell line within the limit of manuscript revision.

3.In Figure 2, LDN193189 augments the growth inhibitory effects of carboplatin (CBDCA). Although LDN193189 reverses the tumor promoting effects induced by exogenous BMP2, LDN193189 is not specific to BMP2 mediated Smad4/5/8 activation, other BMPs, such as BMP4, mediated Smad4/5/8 activation can also be inhibited by this antagonist. Thus, in figure 2H and I, observed effects may not only due to inhibition of BMP2- this needs to be clarified. Do Ishikawa cells express/secrete other BMP members?

Response

As LDN193189 is a BMP type 1 receptor kinase inhibitor, it should inhibit BMP signaling other than BMP2. That is why we performed a sphere formation assay in the presence and the absence of BMP2 (Figure 2I, S3). We could not elucidate how LDN193189 augmented the effect of CBDCA. As you suggested, LDN193189 may potentiate CBDCA’s effect by suppressing BMP signaling other than BMP2. However, it is impossible to assess the effects of different BMP ligands separately because they share common receptors.

We also confirmed mRNA expressions of BMP2, BMP4, BMP6, BMP7, BMP8A and BMP8B by RNA-sequencing of Ishikawa cells (data not shown). On the other hand, Ishikawa cells did not express GDF2 and BMP10 mRNAs. Considering that Ishikawa cells express various BMP ligands, receptor kinase inhibition by LDN193189 is sufficient enough to block BMP signaling.  

4.Title: Tumor promoting effect of BMP signaling in endometrial cancer. The data presented does not support that BMPs promote tumour growth as in vivo studies were not included. Consider changing the title to reflect the key outcomes.

Response

It is true that we have not done any in vivo experiments. However, analyses of clinical datasets suggest tumor promoting effect of BMP signaling proved by in vitro experiments. Therefore, please forgive us to use the title.

5.L33 “an increasing incidence partly due to increased incidence of obesity and increased life-span.” Consider re-wording this sentence.

Response

We modified the sentence.

Please see page 1 line 33 of the revised manuscript.

6.L74: qPCR in one cell line with only selected genes from the family does not ‘validate’ the clinical findings. Please correct this sentence.

Response

We corrected the sentence,

Please see page 2 lines 80–81 of the revised manuscript.

7.Figure 2 legend- were qPCR results normalised to housekeeping genes? This needs to be stated. The authors could also consider using CTRL for control in the qPCR graphs instead of CT, so as not to be confused with cycle threshold Ct (e.g. ‘shown as fold change relative to CT’ in the legend).

Response

All qPCR results were normalized to GAPDH as mentioned in the Materials and Methods section. The description was modified for clarity.

Please see Figure 2 legend line 143 of the revised manuscript.

8.There is no indication of the vehicle control used for experiments involving Imatinib LDN193189 and carboplatin i.e. what are the stocks dissolved in and were these used as vehicle control (where applicable).

.

Response

Imatinib and carboplatin were dissolved in water, whereas LDN193189 was in DMSO. We added this information to the Materials and Methods’ section. Unfortunately, we did not use DMSO as a vehicle control because final concentration of DMSO in LDN193189 was 0.002% in medium.

Please see page 8 lines 278–279 of the revised manuscript.

9.L337 ‘The significance of differences between more than three samples was analyzed by one-way ANOVA’ – should this read ‘three or more samples’?

Response

Yes. We modified the description.

Please see page 10 line 361 of the revised manuscript.

  1. 10. How were the different doses of BMP2 (20ng/ml) and BMP7 (50 ng/ml) selected?

Response

Usually, BMP7 is used at higher concentration compared with BMP2. We validated these concentrations by ID1 induction as shown in Figure 5E.

11.How were the ELISA results normalised? Cell number at confluency varies across different cell types so the experiments should be normalised e.g. to total protein in lysates

Response

We apologize that we did not normalize BMP2 concentration. Therefore, we normalized BMP2 concentration to 1 mg total protein in lysates and modified the panel.

Please see Figure 1G and Figure 1 legend of the revised manuscript.

Reviewer 2 Report

In their study, Fukuda and co-investigators analyzed the effects of BMPs on endometrial and ovarian cancer cell-lines. They reported that „the expression of mRNA for BMP ligants and receptors was found to be frequently increased in EC…”. The promoting effect of BMP on EC has been reversed by LDN193189, and to some extent, by TWSG1. Moreover, BMP2 significantly enhanced the cell sphere formation and migration of EC, as well as „ enhanced EC cell stemness in a c-KIT-dependent manner.”. LDN193189 also augments the growth inhibitory effects of CBDCA. Finally, the Authors concluded that „… BMP signaling is frequently activated in EC patients, combinational treatment with CBDCA and a BMP signaling inhibitor could be beneficial in the treatment of EC patients.”.

This experimental study in nice prepared and presented and it is worth publishing, but before the final acceptance, a revision is recommended. This research is an international collaborational study /Sweden, Japan/.

  1. Why the Authors described the endometrial carcinosarcomas in the introduction and discussion sections? Did the experiments were performed on these rare tumors or cell-lines?
  2. Please briefly discuss the clinical data of BMP signaling alterations in different hormonally-dependent tumors, for example in ovarian cancers or breast cancers. Did the results collected were similar to data currently presented?
  3. How the application of BMP inhibitor could be beneficial in the clinical treatment of EC patients by presenting generally in vitro studies?
  4. Check the reference list, for example references numer 14,32.

Author Response

Details of response to reviewers:

 We highly appreciate the editor and the reviewers for giving us the opportunity to revise our manuscript. We have repeated the comments of the reviewers and have responded to the comments. The page and line numbers refer to our revised manuscript.

Reviewer 2

Comments to the Author

1.Why the Authors described the endometrial carcinosarcomas in the introduction and discussion sections? Did the experiments were performed on these rare tumors or cell-lines?

Response

We described about endometrial carcinosarcoma because BMP signaling could induce sarcomatous components by triggering EMT during its development. However, this is just speculation from the results that BMP2 induced EMT in Ishikawa EC cells and we have not performed experiments with endometrial carcinosarcoma cell lines.

2.Please briefly discuss the clinical data of BMP signaling alterations in different hormonally-dependent tumors, for example in ovarian cancers or breast cancers. Did the results collected were similar to data currently presented?

Response

Overexpression of BMP ligands and receptors was also found in ovarian cancer. We added these findings in the discussion section.

Please see pages 7-8 lines 234–238 of the revised manuscript.

3.How the application of BMP inhibitor could be beneficial in the clinical treatment of EC patients by presenting generally in vitro studies?

Response

EC has a relatively good prognosis. However, recurrence with sarcomatous components is difficult to be cured due to resistance to chemotherapy. BMP inhibitors may be beneficial in preventing such kind of recurrence by suppressing EMT.

  1. 4. Check the reference list, for example references numer 14,32.

Response

We thoroughly checked the reference list. Unfortunately, we could not find evident problems in the list.
